# Boosted Genomic Literacy in Nursing Students via Standardized-Patient Clinical Simulation: A Mixed-Methods Study

**DOI:** 10.3390/nursrep15080297

**Published:** 2025-08-13

**Authors:** Daniel Garcia-Gutiérrez, Estel·la Ramírez-Baraldes, Maria Orera, Verónica Seidel, Carmen Martínez, Cristina García-Salido

**Affiliations:** 1Departamento de Enfermería, Facultad de Ciencias de la Salud de Manresa, Universitat de Vic-Universitat Central de Catalunya (UVic-UCC), Av. Universitària, 4-6, 08242 Manresa, Spain; dgarcia04@umanresa.cat (D.G.-G.); cgarcia@umanresa.cat (C.G.-S.); 2Grupo de Investigación en Simulación e Innovación Transformativa (GRIST), Universitat de Vic-Universitat Central de Catalunya (UVic-UCC), Av. Universitària, 4-6, 08242 Manresa, Spain; 3Instituto de Investigación e Innovación en Ciencias de la Vida y de la Salud de la Cataluña Central (Iris-CC), 08500 Vic, Spain; 4Unitat de Cures Intensives, Althaia Xarxa Assistencial Universitària de Manresa, 08243 Manresa, Spain; 5Servicio de Pediatría, Hospital General Universitario Gregorio Marañón, 28007 Madrid, Spain; morera@ucm.es (M.O.); veronicaadriana.seidel@salud.madrid.org (V.S.); 6Departamento de Biología Celular e Histología, Facultad de Medicina, Universidad Complutense de Madrid, 28040 Madrid, Spain; cmmora@bio.ucm.es

**Keywords:** genetic counseling, clinical simulation, nursing education, mixed-methods study, undergraduate nursing

## Abstract

**Background**: Genomic information is becoming integral to nursing practice, yet undergraduate curricula often provide limited opportunities to apply this knowledge in realistic settings. **Objective**: To evaluate the impact of a clinical simulation-based intervention on nursing students’ learning of genetic counseling, with a focus on knowledge acquisition, communication skills, and student satisfaction. **Methods**: A sequential mixed-methods study was conducted with 30 third-year nursing students enrolled in the elective Genetics Applied to Health Sciences. Quantitative data comprised (i) pre-/post-simulation knowledge tests, (ii) a satisfaction questionnaire, and (iii) final course grades, which were compared with grades of a cohort from the previous academic year that had no simulation component (n = 28). Qualitative insights were gathered through field notes and semi-structured interviews with six purposively selected participants. During the intervention each student rotated through the roles of genetic-counseling nurse, patient, and observer, followed by a facilitated debriefing. **Results**: Post-simulation knowledge scores and final course grades were significantly higher than both baseline values and the historical comparison cohort. Students reported very high satisfaction, highlighting the authenticity of the scenarios and the usefulness of immediate feedback. Qualitative analysis showed that role rotation fostered deeper understanding of counseling complexities, improved empathic communication, and bolstered self-confidence when discussing hereditary risk. **Conclusions**: Embedding standardized-patient simulation into undergraduate genetics courses measurably improves students’ knowledge, communication proficiency, and satisfaction. These findings support incorporating similar simulation-based learning activities to bridge the gap between theoretical genetics content and real-world nursing practice.

## 1. Introduction

Genetics knowledge is increasingly essential in healthcare, particularly for nursing professionals. It enables them to assess genetic risks, support patients with hereditary conditions, and contribute to preventive care strategies [1]. For decades, nurses have played a pivotal role in identifying genetic-based health issues in patients and their families [2]. The evolution of genetics in nursing has led to more comprehensive and personalized care, encompassing not only medical interventions but also emotional support, education, and care coordination [3].

In Spain, strategic shifts in primary care have emphasized early diagnosis to improve health problem management. However, genetics education within nursing curricula remains inconsistent across countries. While the USA and UK have pioneered specific programs for genetic counselors, this professional role is not yet officially recognized in Spain. Despite the inclusion of some genetics-related content in Spanish undergraduate health sciences programs, there remains a considerable gap, particularly in genetic counseling training. Addressing this gap is crucial for equipping nursing professionals with the necessary knowledge and skills to participate in multidisciplinary teams involved in research, gene therapy development, and the monitoring and prevention of genetic diseases. Consequently, integrating genetics education into nursing curricula is imperative to enhance patient care and advance the field of healthcare [1,2,3,4,5,6].

A comprehensive analysis conducted in Spain, reviewing 4720 course syllabi from 118 Spanish universities, highlighted substantial inconsistencies in genetics and genomics education, demonstrating the lack of a standardized curriculum in this field [7]. Notably, 12 institutions did not include any genetic or genomic content, while among those that did, 43% focused exclusively on basic genetics, and only 57% covered advanced topics such as genetic counseling and pharmacogenetics. Furthermore, traditional lectures and seminars remain the predominant teaching methodologies, with limited opportunities for practical training. On average, undergraduate nursing programs in Spain allocate only 3 to 6 ECTS credits to genetics and genomics within a total of 240 credits. These findings highlight the urgent need to update and standardize curricula to ensure comprehensive education in these critical areas, ultimately preparing nursing professionals for the evolving demands of modern healthcare [7].

Clinical simulation has emerged as a powerful educational tool that allows nursing students to engage in real-life scenarios, improving their critical thinking, decision making, and patient interaction skills while minimizing clinical errors [8]. This methodology consists of three phases: briefing, simulation, and debriefing, enabling participants to apply their skills, knowledge, and attitudes by integrating theoretical learning with practical application. Recent studies confirm that high-fidelity simulation significantly enhances not only technical performance but also communication, teamwork, and non-technical skills among nursing students and professionals [9,10,11,12,13]. These improvements are particularly relevant in the context of genetic counseling, where effective communication, decision making, and empathy are essential. Simulation-based education has been shown to improve students’ clinical reasoning, critical thinking, and preparedness for real-life practice, offering a safe environment for experiential learning and reflective debriefing. Repeated exposure to high-fidelity simulation not only consolidates learning but also increases confidence and readiness for clinical challenges. This body of evidence supports the integration of high-fidelity simulation in nursing curricula as a pedagogical tool to bridge the gap between theoretical knowledge and clinical application, especially in complex areas like genetics.

Various studies have identified deficiencies in genetic and genomic literacy among health sciences students worldwide, reinforcing the necessity for updated educational programs [14,15,16,17,18,19]. In Spain, despite curriculum variability, there are no specific studies evaluating the integration of genetics and genomics education in nursing programs.

The significance of genetics in personalized care and disease prevention is increasingly recognized worldwide. Internationally, genetic nurses and counselors play crucial roles in obtaining family histories, recommending screening programs, requesting predictive genetic studies, providing genetic counseling, offering emotional support, and guiding self-care. These findings underscore the necessity of incorporating genetics and genomics education into nursing curricula to prepare future professionals for the evolving healthcare landscape [20,21,22].

Strengthening the training of future nursing professionals in Spain by embedding genetics as a fundamental component of education is crucial to ensuring they can effectively navigate the growing role of genomics in healthcare. In recent years, there has been growing interest in utilizing clinical simulation for genetic counseling education in nursing schools. This project was initiated following a pedagogical inquiry into the implementation of clinical simulation sessions in the elective course “Applied Genetics in Health Sciences” at the Universidad Complutense de Madrid. The research aims to answer the following question:

What impact does the implementation of clinical simulation have on student learning about genetic counseling?

To address this, the study analyzes the effects of clinical simulation on university nursing students’ learning, focusing on knowledge acquisition, communication skill development, and student satisfaction.

## 2. Materials and Methods

### 2.1. Study Design

The primary objective of this study was to evaluate the impact of a clinical simulation-based educational intervention on nursing students’ knowledge acquisition related to genetic counseling. Secondary objectives included assessing potential improvements in communication skills and student satisfaction with the learning experience. Given the exploratory nature and mixed-methods approach of the study, no formal hypotheses were established. However, improvements in these areas were anticipated based on prior literature.

A quasi-experimental pre–post design was used. The intervention group (academic year 2023–2024) was evaluated before and after participating in the simulation-based training. In addition, a historical cohort (academic year 2022–2023) that received traditional instruction without simulation was used as a non-equivalent control group, enabling both within-group and between-group comparisons of knowledge acquisition.

The study adopted a sequential mixed-methods design, combining quantitative and qualitative methodologies [23] to enhance both depth of understanding and methodological robustness. Quantitatively, the study aimed to measure learning outcomes and student satisfaction. Qualitatively, it was grounded in a constructivist–interpretative paradigm and informed by a hermeneutic phenomenological approach, which seeks to explore the meanings and intentions behind human actions from the participants’ own perspectives [24,25,26].

### 2.2. Participants

The study was carried out within the transversal elective course “Applied Genetics in Health Sciences” offered to students of the nursing, physiotherapy, and podiatry degrees of the Universidad Complutense de Madrid and taught during the 2023–2024 academic year.

The total number of students officially enrolled in the course was 30, and all of them participated in the intervention. Participation was voluntary, and inclusion required attendance at the clinical simulation sessions and written informed consent. Since no student declined participation or failed to meet these basic conditions, the entire enrolled cohort (n = 30) was included in the study. No exclusion criteria were applied.

Because the entire cohort of 30 enrolled students was included, an a priori sample-size calculation was not feasible; however, a post hoc power analysis (Cohen’s *d* ≈ 2.8) yielded > 99% power at α = 0.05, confirming that the census sample was adequate for the primary outcome.

### 2.3. Simulation Design

Before the implementation of the clinical simulation, the elective course “Applied Genetics in Health Sciences” was taught using traditional learning methods. These included face-to-face lectures, theoretical seminars, and the analysis of clinical cases without any practical simulation component. This conventional approach provided students with conceptual knowledge but limited opportunities for experiential or skills-based learning.

Clinical simulation sessions were designed to replicate common scenarios in a genetic counseling clinic, focusing on breast cancer, Klinefelter syndrome, and cystic fibrosis. Three distinct simulation cases were developed by the authors, following pedagogical principles established in the literature on clinical simulation and reflective practice [18,19,20]. The scenarios were designed to foster deep learning through experiential engagement, feedback, and structured reflection.

The development process integrated theoretical frameworks on deliberate practice and debriefing methodology, particularly the “Debriefing with Good Judgment” model [20] and guidance on promoting clinical reasoning through simulation [18]. Each session included structured phases—briefing, simulation, and debriefing—aligned with current best practices in simulation-based education.

Students were divided into subgroups of five and rotated through roles in each scenario:-Nurse (acting as genetic counselor): Students practiced delivering genetic counseling to patients.-Patient: Students acted as patients receiving genetic counseling.-Observer: Students observed interactions, noting communication styles and clinical skills.

Each student experienced all three roles across the scenarios to gain multiple perspectives. Each simulation was facilitated by a subject expert and a simulation facilitator.

The full clinical simulation session lasted five hours (300 min) and followed a carefully structured sequence:-General briefing (30 min): A shared session introducing the methodology, objectives, structure, and content of the three clinical cases.-Three simulation stations, each structured as follows:○Briefing (10 min) tailored to the specific scenario.○Simulation (20 min) with standardized patients.○Debriefing (30 min) focused on reflection and learning.-Transitions between scenarios (2 × 10 min = 20 min).-Mid-session break (30 min) to allow for rest and decompression.-Final meta-debriefing (40 min): A concluding group session to integrate the learning outcomes across all scenarios and roles.

This time management structure ensured that students could engage deeply in each phase of the simulation while benefiting from guided reflection, practical application, and peer-to-peer learning.

### 2.4. Study Variables and Measurement Instruments

The main variables investigated in this study, along with the instruments and procedures used to assess them, were as follows:-Knowledge Acquisition: This was evaluated through multiple-choice knowledge tests designed specifically for the course. Each test included 10 questions covering both theoretical and practical aspects of genetic counseling. These tests were administered immediately before (pre-simulation) and after (post-simulation) the simulation sessions to measure learning progression.-Communication Skills: These were assessed through qualitative analysis of students’ interactions during the simulation, as observed and recorded by facilitators. A thematic analysis was conducted using Atlas.ti 9 version 23 software, focusing on communication effectiveness, empathy, and clarity in delivering complex information. In addition, semi-structured interviews (see Appendix A Table A1) were carried out with a sample of participants to explore perceptions regarding their communication skill development.-Student Satisfaction: This was measured using the validated satisfaction questionnaire developed by Durá Ros [27], administered at the end of the simulation. The questionnaire uses a Likert scale ranging from 1 (very dissatisfied) to 5 (very satisfied) and evaluates simulation quality, learning usefulness, and perceived confidence improvement. The instrument is an adaptation of validated scales developed by Durá Ros and has demonstrated strong psychometric properties. In prior applications, the questionnaire achieved a Cronbach’s alpha coefficient of α = 0.87, indicating high internal consistency and reliability in measuring student satisfaction with simulation-based learning.

### 2.5. Data Collection

This study used a mixed-methods approach to evaluate the impact of clinical simulation on nursing students.

Quantitative data were collected through paper-based questionnaires administered immediately before and after the simulation sessions. These instruments assessed students’ knowledge acquisition, communication skills, and overall satisfaction.

Qualitative data were gathered through open-ended written reflections submitted at the end of the session and semi-structured group interviews with a subsample of students. The interviews, conducted by an external researcher not involved in the teaching process, explored learners’ experiences, perceptions of realism, and the perceived impact on their professional development. All qualitative responses were transcribed and analyzed using thematic content analysis.

### 2.6. Data Analysis

#### 2.6.1. Quantitative Data Analysis

Quantitative analyses were performed in IBM SPSS Statistics v27. Within the 2023–24 intervention cohort, pre- and post-simulation knowledge scores were compared with paired *t*-tests, and effect sizes were expressed as Cohen’s *d*. To contextualize these results, final course grades were contrasted with those of a historical cohort from 2022–2023 (which received traditional instruction without simulation) using independent-samples *t*-tests. Student-satisfaction items were summarized descriptively as means and standard deviations. Assumptions for parametric testing were verified with Shapiro–Wilk tests of normality and Levene’s test for homogeneity of variances, all of which were satisfied (*p* > 0.05). Statistical significance was set at α = 0.05 (two-tailed).

#### 2.6.2. Qualitative Data Analysis

Participants for the qualitative phase were selected through purposive sampling, ensuring variation in academic performance and engagement levels during the simulation. All students were informed of the research purpose prior to data collection and voluntarily agreed to be interviewed. Inclusion criteria included full participation in the simulation sessions and post-session debriefings. No exclusion criteria were applied.

Interviews were transcribed verbatim and, along with field notes, analyzed using thematic analysis with Atlas.ti 9 software. The process involved coding, categorizing, and identifying themes related to learning experiences, skill development, and perceptions of the simulation’s effectiveness [28]. Findings from the qualitative analysis were used to contextualize and enrich the quantitative results [23,29].

Data saturation was achieved after six interviews, as no new themes or insights emerged from the additional data. Data saturation occurs when additional data collection fails to produce new information relevant to the research questions, indicating that the sample size is sufficient to explore the phenomena under study [30]. Achieving data saturation enhances the credibility and trustworthiness of qualitative research findings.

## 3. Results

The acceptance of simulation methodology in the taught course was excellent, with all registered students (n = 30) participating in the simulation. Most participants were female, comprising 80.8% (24 females vs. 6 males).

### 3.1. Knowledge Acquisition

Results from the pre-simulation knowledge test showed a mean score of 3.6 (SD: 2.19) out of 10. After the simulation, the post-simulation test results displayed a significantly higher mean score of 9.2 (SD: 1.12) out of 10. A paired samples *t*-test was conducted to compare the pre- and post-simulation scores. The test revealed a statistically significant improvement in scores (t (29) = −13.32, *p* < 0.001), indicating that the simulation had a substantial positive effect on students’ knowledge acquisition (Figure 1).

### 3.2. Final Course Grade Comparison

We analyzed the final exam results of students from the previous academic year (2022–2023), which did not include clinical simulation as a supplementary teaching method, and compared them with those from the current academic year (2023–2024), which did include simulation. In the 2022–2023 academic year, 28 students were assessed, achieving an average score of 7.08 (SD: 1.22) out of 10. In contrast, in the 2023–2024 academic year, 30 students achieved a higher average score of 8.26 (SD: 0.84) out of 10. An independent samples *t*-test was performed to compare the final grades between the two cohorts. The results showed a statistically significant difference in scores (t (56) = −4.17, *p* < 0.001), suggesting that the implementation of simulation contributed to improved academic performance (Figure 2). Both cohorts completed the same final examination, using an identical structure, content, and scoring criteria, which ensures the comparability and validity of the results.

### 3.3. Student Satisfaction

Regarding the quality and satisfaction survey, the average satisfaction score was 4.56 out of 5 for the genetic clinical simulation conducted. When analyzing responses by gender, females had an average satisfaction score of 4.54 (SD: 0.51), while males had an average score of 4.64 (SD: 0.48). An independent samples *t*-test indicated that there was no significant difference in satisfaction scores between genders (t (28) = −0.45, *p* = 0.65).

### 3.4. Qualitative Analysis

According to the survey, students were invited to voluntarily participate in interviews. From the information gathered from individual interviews and field notes, a qualitative analysis was conducted to gain a deeper understanding of students’ experiences with the implementation of genetic simulation.

A total of six students were interviewed. Upon initial reading, organization, and classification of data using basic coding units, 263 units of meaning were generated [28]. Subsequently, an initial open coding [29] was performed, initially grouping the units of meaning into 12 categories, followed by axial coding [30,31]. From these categories, four metacategories emerged: learning, attitude, feedback, and roles (see Appendix A Table A2)

At the final stage of analysis, the data were organized using selective coding [30,31] into two domains or emerging thematic cores: dimensions of clinical learning and comprehensive evaluation in clinical simulation (see Appendix A Table A3), and the categories, metacategories, and research dimensions or thematic cores (see Appendix A Table A4).

## 4. Discussion

This study examines the impact of clinical simulation in nursing education, focusing on genetic counseling. The research utilized a mixed-methods approach, combining quantitative and qualitative data to provide a comprehensive understanding of the simulation’s effects.

Quantitative results showed improved post-simulation assessment scores and higher final grades in the academic year that implemented simulation compared to the previous year without it. Qualitatively, students and educators observed effective transfer of theoretical and practical learning to simulated clinical environments.

The simulation provided a safe, non-punitive learning environment that allowed for controlled repetition and error. Students reported high satisfaction levels with the clinical simulation as a teaching tool, which was measured using the validated Durá Ros scale.

This pioneering study in implementing simulation for genetics education in nursing demonstrates its effectiveness in enhancing students’ knowledge, skills, and satisfaction. The research underscores the potential of simulation as an innovative teaching methodology in genetic counseling education, aligning with previous studies that highlight simulation’s advantages in healthcare professional training [21,22,32,33,34,35,36,37,38,39,40,41].

Recent evidence supports that high-fidelity simulation improves clinical performance, confidence, communication, and decision-making skills among undergraduate nursing students, contributing to better preparedness for real clinical environments [11,12,13]. Moreover, systematic reviews confirm its value in developing non-technical competencies such as empathy, collaboration, and leadership, which are essential in genetic counseling scenarios [9,10].

Regarding the specific objective of exploring the experiential learning of clinical simulation, results are discussed based on identified metacategories.

Within the “learning” metacategory, participants expressed: -“Clinical simulation has allowed me to acquire solid theoretical genetic knowledge on different medical conditions”.-“It’s group learning that develops knowledge… I’ve also developed essential practical skills for my future nursing practice thanks to simulation”.-“Simulation has improved my ability to effectively diagnose genetic diseases”.

Debriefing highlighted the importance of team case discussion, confirming the significance of group learning in acquiring necessary knowledge and skills essential for professional roles in healthcare and specifically genetic counseling. This finding aligns with previous studies emphasizing clinical simulation as an effective technique for skill development related to patient care [42,43,44]. 

These findings regarding knowledge acquisition and student satisfaction are consistent with previous studies that highlight the effectiveness of high-fidelity simulation in improving theoretical understanding and learner engagement in nursing education. For example, Vangone et al. [10] reported improved cognitive and affective outcomes among healthcare professionals exposed to simulation-based training focused on non-technical skills. Similarly, Adell-Lleixà et al. [9] demonstrated significant gains in satisfaction and communication competencies among intermediate care nurses following ultra-realistic clinical simulation sessions. Our results build on this evidence by confirming similar benefits in the specific and emotionally nuanced context of genetic counseling, suggesting the adaptability and pedagogical value of simulation in highly specialized domains.

In the “attitudinal” metacategory, participants expressed a positive perception of clinical simulation, reflecting its value in enhancing confidence and security in managing real patients. These results are consistent with scientific evidence suggesting that clinical simulation enhances confidence and security among students in facing future clinical scenarios. Several examples include: -“It allows us to lose the fear of treating patients… helps us build confidence to gather information from the patient”.-“Simulation boosts my confidence in dealing with real patients as it familiarizes us with genetic counseling consultations”.-“Practice in simulation helps us overcome the fear of making mistakes… it makes us feel secure in multidisciplinary work”.

Participants recognized the importance of assertive communication in clinical practice and valued simulation as a tool for developing these skills. This finding is in line with evidence indicating that simulation fosters greater confidence in interdisciplinary and patient-centered communication skills [45,46].

Although students reported increased satisfaction, confidence, and skill acquisition, further research is needed to determine whether these improvements are sustained over time and how effectively they are applied in real clinical environments. Longitudinal designs and multi-institutional approaches would offer a broader perspective on these outcomes and their generalizability. Although the qualitative findings in this study align with previous research highlighting the value of simulation in fostering communication skills, confidence, and empathy among nursing students [9,10,11,12,13,45,46], this study contributes additional nuance by applying simulation specifically to genetic counseling contexts. In contrast to many prior studies that focus on general clinical or emergency scenarios, our results reveal that the emotional complexity and sensitivity required in genetic counseling provoke deeper self-reflection and role-based learning. For instance, while other studies emphasize the role of debriefing or technical competence [36,47], our participants highlighted the transformative effect of rotating through patient and observer roles, which is less commonly explored. This suggests that simulation scenarios involving emotionally charged and ethically complex content may offer distinct pedagogical benefits that merit further investigation.

In the “feedback” metacategory, various perceptions were reflected on the relationship between simulation practice and theoretical knowledge, highlighting its value as reinforcement and complement to academic teaching. Some statements include:-“Simulation helps us gain an idea and vision of what it’s like in reality”.-“Sometimes simulation practice does not align with theory”.-“Simulation practice doesn’t reflect reality”.

Clinical fidelity in simulation refers to credibility or resemblance to real-life experiences. Achieving high fidelity is challenging, as evidenced in previous studies where accurately replicating real situations and ensuring participants react similarly is difficult [47,48]. Participant opinions varied on the relationship between simulation practice and theoretical knowledge, underscoring its value as reinforcement and complement to academic teaching. 

In the “role” metacategory, this study represents a significant contribution to nursing education by exploring the impact of clinical simulation methodology, involving active student participation in three fundamental roles: nurse, patient, and observer. Findings revealed promising results highlighting the effectiveness and implications of this multifaceted approach. Specifically:-Direct experience from the patient’s perspective significantly improved students’ understanding and application of nursing care principles, fostering deeper empathy and holistic understanding of patient needs. Previous scientific evidence has described empathy as crucial in both observer and nurse roles, but not previously in the patient role [49].-The observer role provided students with opportunities to reflect on their own nursing practices and behaviors, identifying areas for improvement and strengthening critical self-assessment.

For example:-“Simulation is a valuable tool… I’ve improved the application of concepts in clinical practice, especially after being the patient and considering how I would better understand genetic counseling advice”.-“When observing peers during simulations, you think of many ways to lead the situation… whether they do it well or if there could be alternative diagnoses… which helps organize genetic counseling”.-“The experience as a patient in simulations provides a more comprehensive understanding of the clinical environment”.-“As a nursing student, when I played the patient role in simulations, it prompted me to consider issues I’ll have to address in the future that I hadn’t previously considered… it made me think differently… I enjoyed experiencing being both the patient and the nurse”.

Simultaneously, it was observed that students experiencing different roles detected that clinical simulation in genetic counseling can enhance understanding of professional practice and strengthen ethical decision-making skills. This approach, supported by previous evidence, offers an exceptional opportunity for risk-free learning, enabling participants to experience and reflect on complex situations without the potential to harm real patients [47,50].

Furthermore, participants recognized simulation as an effective tool for assessing genetic counseling performance and learning through error identification. Through logical and deductive reasoning processes, students develop knowledge that surpasses mere memorization, allowing for deep understanding. This type of learning enriched by clinical simulation is characterized by meaning, understanding, retention, and transferability, all essential for the comprehensive training of future nursing professionals [32,51,52,53,54]. For instance:-“Acting as an observer in simulations has allowed me to develop critical and reflective skills crucial for my future as a nurse”.-“The observer role has enabled me to analyze genetic counseling objectively and reflect on it”.

Finally, integrating the three roles in nursing education through clinical simulation enriches the student’s educational experience, facilitating holistic development of clinical and caregiving skills necessary to tackle future challenges in professional practice. This methodology provides a safe and structured context where students can refine communication skills, manage emotional situations, and make patient-centered decisions related to genetic counseling, thus promoting evidence-based practice and patient-centered care.

Furthermore, simulation offers a platform for identifying and rectifying errors without adverse consequences for real patients, fostering acquisition and consolidation of essential knowledge for excellence in clinical care. Addressing this need for adequate training not only benefits individuals and families affected by genetic conditions but also strengthens and elevates the nursing profession.

This study represents an exploratory contribution to the integration of clinical simulation in genetics education for nursing. While the findings are promising, they should be interpreted within the context of a single-center, cross-sectional design. Future multicenter and longitudinal studies are essential to validate these initial results and examine the long-term retention of genomic knowledge and the transferability of communication skills to real clinical settings.

### Limitations

Despite the promising findings, this study has some limitations that should be acknowledged to contextualize its contributions and guide future research:Sample Size and Scope: Conducted with 30 students from a single institution, the sample limits generalizability. However, this size is appropriate for an exploratory design and was complemented by a mixed-methods approach to enhance internal validity.Non-Randomized Design: No randomized control group was included, limiting causal inference. Nonetheless, a quasi-experimental comparison was made with the previous academic year, offering a reasonable benchmark. The decision not to randomize was based on ethical and pedagogical considerations.Qualitative Sampling Bias: Interviews were conducted with students who had prior simulation experience, which may have biased responses positively. To mitigate this, data triangulation was used with field notes and observations from facilitators to broaden perspectives.Measurement and Bias Risk: Learning gains and satisfaction were assessed with validated knowledge tests and self-reported questionnaires, which may be subject to social-desirability or novelty effects; triangulation with qualitative data and historical-cohort comparison helped mitigate this risk. Because examination booklets were retrieved immediately after each sitting and a parallel-form test was applied to the 2023–2024 cohort, the likelihood of item contamination between academic years was greatly reduced, although informal sharing of questions cannot be completely ruled out. This potential test-transmission bias is therefore acknowledged when interpreting the grade comparison.Short-Term Assessment: Outcomes were evaluated immediately post-intervention, without long-term follow-up. Future studies should include delayed assessments to measure retention and application of skills in clinical practice.

## 5. Conclusions

Clinical simulation in genetic counseling education for nursing students has demonstrated significant potential as an effective teaching tool. This study highlights improvements in students’ knowledge acquisition, communication skills, and self-confidence through immersive role-playing as genetic counselors, patients, and observers.

By fostering active learning and clinical communication, this approach enhances student engagement and contributes to the development of patient-centered care competencies. The findings reinforce the importance of innovative educational strategies that bridge the gap between theoretical genetics education and its practical application in clinical settings.

Given the growing role of genomics in healthcare, integrating clinical simulation into nursing curricula represents a valuable pedagogical advancement that could improve patient outcomes by enhancing healthcare providers’ communication skills and preparedness for genetic counseling scenarios.

Further research should explore its long-term educational impact, particularly in terms of knowledge retention and real-world application. The expansion of this methodology to other healthcare education settings could strengthen interprofessional collaboration and improve the quality of genetic counseling services provided by future nursing professionals.

## Figures and Tables

**Figure 1 nursrep-15-00297-f001:**
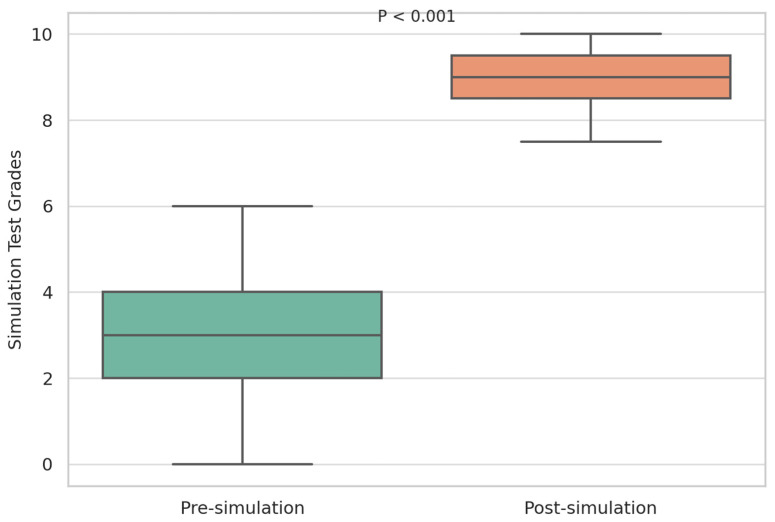
Positive effect on students’ knowledge acquisition.

**Figure 2 nursrep-15-00297-f002:**
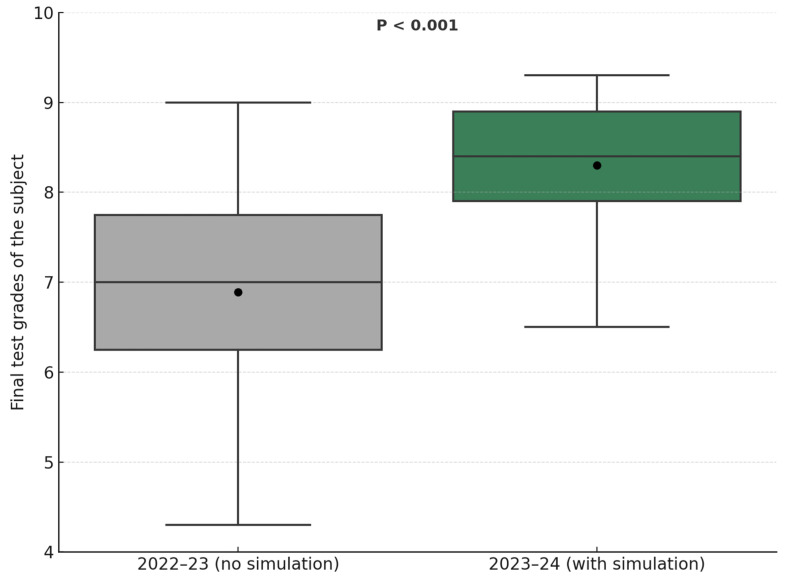
The implementation of simulation contributed to improved academic performance.

## Data Availability

The data presented in this study are available on reasonable request from the corresponding author. Due to privacy and ethical restrictions, the data are not publicly accessible. Access will be provided in a manner that ensures the confidentiality of the participants and complies with ethical guidelines.

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
