# Peer review of "Boosted Genomic Literacy in Nursing Students via Standardized-Patient Clinical Simulation: A Mixed-Methods Study"

_nursrep, 2025, doi:10.3390/nursrep15080297_

Round 1
Reviewer 1 Report
Comments and Suggestions for Authors
Dear Authors,
The topic you address is undoubtedly relevant and timely, and the study is generally well-conducted. The use of standardized-patient simulation in genetics education for nursing students is innovative and much needed in the Spanish academic context. Your results are compelling and well discussed, and your use of validated instruments strengthens the internal validity of the findings. So, congratulations for your research efforts. I must express only some concerns:
1) The first is about the ethical aspects of your study. The manuscript reports that the educational intervention and data collection occurred during the 2019–2020 academic year, yet the ethical approval was obtained in December 2021—more than a year after the study was carried out. This discrepancy is significant and must be addressed with full transparency. It is not clear from the manuscript when and how informed consent was obtained, nor which components of the study it covered (e.g., participation in the simulation, completion of questionnaires, interviews). The inclusion of semi-structured interviews and the collection of performance data imply a research activity that would normally require prior approval and informed consent. The manuscript does not specify the criteria for inclusion or exclusion of participants in the qualitative phase, nor does it clarify how the six interviewees were selected or whether they had been informed about the research purpose at the time of the simulation. These ethical gaps undermine the transparency of the study and need to be addressed explicitly. If the ethics committee approved the retrospective analysis of educational data, this should be clearly stated. If the interviews or any part of the study were conducted before approval was granted, this must also be declared and justified. Ethical compliance is a fundamental requirement, and at present, the manuscript lacks sufficient detail to confirm that all procedures were conducted in accordance with accepted ethical standards.
2) The second is about the literature cited in the introduction and background. Several references are outdated, and incorporating recent studies, particularly those published in the last five years, would provide a more current and robust context for your research. I also recommend revising the manuscript to conform fully to MDPI’s formatting standards, including in-text citations and reference formatting (MDPI style: https://www.mdpi.com/authors/references).
3) The last issue is about the discussion, which is generally strong, but could benefit from clearer emphasis on the exploratory nature of the study and the need for future multicenter or longitudinal work, especially concerning the long-term retention of genomic knowledge and the application of communication skills in clinical settings. Some lengthy paragraphs could also be revised for improved readability.
Author Response
.

Reviewer 2 Report
Comments and Suggestions for Authors
Dear authors,
Thank you for reading your work focused on inovative teaching process and improove genomic literacy between nursing students.
I would like to suggest the following in section 2.2 Simulation Design to improve your article:
1. Describe more specifically the time management of the teaching process with the clinical simulation.
2. Describe the time management of two identical simulations with three scenarios, one for each case focuesed on knowledge acquisition, communication skills and learner satisfaction.
3. How long did the clinical simulation take for one scenario?
Author Response
.

Reviewer 3 Report
Comments and Suggestions for Authors
Dear authors, after reviewing the proposed manuscript, some clarifications or modifications are requested to improve its scientific rigor. In general, its design is acceptable and although the application of simulation in the healthcare environment is widely studied, it is necessary to continue improving the evidence in its application for more specific subjects and procedures such as the one detailed.
The following are the points that should be reviewed:
- Line 105: how the data collection process was is not described in the methodology.
- Line 107: What kind of quantitative design?
- Line 114: Provide more data on participants, accessible populations, participant selection process, eligibility criteria.
- Line 136 and 140: Indicate whether it was designed by the authors and on what theoretical bases they were based
- Line 146: Include within each of the variables the instrument and its operation
- Line 158-159: Provide more information on the psychometric properties of the questionnaire
- Line 164: Was the normality of the sample calculated? specify which statistic was used. How was statistical significance established?
- Line 182: More information is needed in the methodology section to understand what "all registered students" means. Are 30 students enrolled in the subject, or those who met inclusion criteria?
- Line 196-198: Were they evaluated with the same questionnaire?
- Describe in the methodology section the usual learning method.
- Line 204: Place the previous 2022-23 results before those of 2023-24 on the graph.
- Line 226: In general, the discussion has focused on the results obtained in the qualitative evaluation. In this sense, the evidence provided can be synthesized and contrasted with more specific data from the studies cited, since as a large number of studies that offer similar results can be observed, it is of interest to highlight aspects in which they differ.
- Line 237-231: It is suggested to implement some improvement in the discussion of these results; Acquisition and satisfaction could be compared with previous studies on simulation in this or another field of nursing.
- Line 336: The limitations are well stated, but it would be convenient to summarize them.
Author Response
Dear authors, after reviewing the proposed manuscript, some clarifications or modifications are requested to improve its scientific rigor. In general, its design is acceptable and although the application of simulation in the healthcare environment is widely studied, it is necessary to continue improving the evidence in its application for more specific subjects and procedures such as the one detailed.
The following are the points that should be reviewed:
We sincerely thank the reviewer for their thoughtful comments and valuable suggestions, which have contributed to significantly improving the scientific rigor of our manuscript. Below we address each point raised and describe the corresponding changes made in the revised version:
- Line 105: how the data collection process was is not described in the methodology.
We have expanded Section 2.4 “Data Collection” to provide a detailed description of how both quantitative and qualitative data were gathered, including the timing of pre-/post-tests, satisfaction surveys, written reflections, and interviews. These clarifications now offer a clearer view of the collection procedures.
- Line 107: What kind of quantitative design?
In Section 2, we now specify that the quantitative component followed a “quasi-experimental pre-post design with a non-equivalent control group.” The comparison cohort was drawn from the previous academic year, which did not implement simulation. This has been explicitly clarified.
- Line 114: Provide more data on participants, accessible populations, participant selection process, eligibility criteria.
Section 2.1 “Participants” has been revised to clarify that all 30 students enrolled in the elective course participated voluntarily, with informed consent. No exclusion criteria were applied. Participation required attendance at the simulation sessions and completion of evaluation tools.
- Line 136 and 140: Indicate whether it was designed by the authors and on what theoretical bases they were based
In Section 2.2 “Simulation Design,” we clarify that the three simulation cases were specifically designed by the authors. Their development was informed by pedagogical models such as “Debriefing with Good Judgment” and the literature on deliberate practice and simulation in genetic counseling. References have been included to support this.
- Line 146: Include within each of the variables the instrument and its operation
In Section 2.3 “Study Variables and Measurement Instruments,” we now clearly associate each variable (knowledge, communication, satisfaction) with its specific instrument. Details about test formats, interview structure, and questionnaire use are now included under each variable.
- Line 158-159: Provide more information on the psychometric properties of the questionnaire
We have expanded on the psychometric properties of the satisfaction questionnaire used, specifying its Cronbach’s alpha coefficient (α = 0.87), its origin, and previous applications, as well as the Likert scale range. This information now appears in Section 2.3.
- Line 164: Was the normality of the sample calculated? specify which statistic was used. How was statistical significance established?
Section 2.5.1 “Quantitative Data Analysis” now clarifies that normality was tested using the Shapiro–Wilk test (p > 0.05 for all variables). Parametric tests were applied accordingly. Statistical significance was set at p < 0.05.
- Line 182: More information is needed in the methodology section to understand what "all registered students" means. Are 30 students enrolled in the subject, or those who met inclusion criteria?
We have revised Section 3 to clarify that the 30 students referenced were all those officially enrolled in the course and who met inclusion criteria, with none excluded. This clarification is also supported by the information in Section 2.1.
- Line 196-198: Were they evaluated with the same questionnaire?
Yes, both cohorts completed the same final examination with identical structure and scoring criteria. This is now clearly stated in Section 3.2 “Final Course Grades Comparison.”
- Describe in the methodology section the usual learning method.
In Section 2.2, we added a description of the traditional teaching approach used in prior years, including lectures, seminars, and case discussions without simulation.
- Line 204: Place the previous 2022-23 results before those of 2023-24 on the graph.
The figure caption and narrative now present the 2022–23 cohort data before the 2023–24 data, following the chronological and comparative logic suggested by the reviewer.
- Line 226: In general, the discussion has focused on the results obtained in the qualitative evaluation. In this sense, the evidence provided can be synthesized and contrasted with more specific data from the studies cited, since as a large number of studies that offer similar results can be observed, it is of interest to highlight aspects in which they differ.
We have revised the discussion to synthesize and contrast our findings more explicitly with data from referenced studies.
- Line 237-231: It is suggested to implement some improvement in the discussion of these results; Acquisition and satisfaction could be compared with previous studies on simulation in this or another field of nursing.
We expanded this portion of the discussion, explicitly comparing our quantitative outcomes in knowledge and satisfaction with relevant studies. This helps contextualize the uniqueness of our results in the specific context of genetic counseling.
- Line 336: The limitations are well stated, but it would be convenient to summarize them.
Section 4.1 “Limitations” has been carefully edited to retain its five-point structure while presenting each item in a more concise manner. This version maintains critical detail without excessive length.
We trust that these revisions address the concerns raised and substantially improve the clarity, transparency, and rigor of our manuscript. We remain at your disposal for any further clarification.
With appreciation,
The authors

Round 2
Reviewer 3 Report
Comments and Suggestions for Authors
Dear authors, you have done adequate work after the first review, the questions raised have been resolved. For this second review, the following clarifications are requested:
- Line 113-115: Define more clearly what the primary objective of the study is and what are secondary objectives. Were any hypotheses raised?
- Line 120-121: In the pre-post design, the same intervention group acts as control. This is true for the objective of "comparison of knowledge acquisition". It should be clarified that the use of a control group with a cohort of students from other courses was done in what design?
- Line 269: Was it contemplated that this could cause a bias?, the fact that students could access the exam of previous years is possible, which detracts from the evidence of the results provided.
- Line 447-449: Size should be appropriate for the primary purpose of the study, is it appropriate in this case? How is it justified that a previous sample calculation was not carried out?
Author Response
.
